# Research on Network Intrusion Detection Model Based on Hybrid Sampling and Deep Learning

**DOI:** 10.3390/s25051578

**Published:** 2025-03-04

**Authors:** Derui Guo, Yufei Xie

**Affiliations:** School of Intelligence Science and Technology, Beijing University of Civil Engineering and Architecture, Beijing 102616, China; 2108540023026@stu.bucea.edu.cn

**Keywords:** network intrusion detection, hybrid sampling methodologies, temporal convolutional networks, residual networks, multi-headed attention mechanisms

## Abstract

This study proposes an enhanced network intrusion detection model, 1D-TCN-ResNet-BiGRU-Multi-Head Attention (TRBMA), aimed at addressing the issues of incomplete learning of temporal features and low accuracy in the classification of malicious traffic found in existing models. The TRBMA model utilizes Temporal Convolutional Networks (TCNs) to improve the ResNet18 architecture and incorporates Bidirectional Gated Recurrent Units (BiGRUs) and Multi-Head Self-Attention mechanisms to enhance the comprehensive learning of temporal features. Additionally, the ResNet network is adapted into a one-dimensional version that is more suitable for processing time-series data, while the AdamW optimizer is employed to improve the convergence speed and generalization ability during model training. Experimental results on the CIC-IDS-2017 dataset indicate that the TRBMA model achieves an accuracy of 98.66% in predicting malicious traffic types, with improvements in precision, recall, and F1-score compared to the baseline model. Furthermore, to address the challenge of low identification rates for malicious traffic types with small sample sizes in unbalanced datasets, this paper introduces TRBMA (BS-OSS), a variant of the TRBMA model that integrates Borderline SMOTE-OSS hybrid sampling. Experimental results demonstrate that this model effectively identifies malicious traffic types with small sample sizes, achieving an overall prediction accuracy of 99.88%, thereby significantly enhancing the performance of the network intrusion detection model.

## 1. Introduction

In recent years, the rapid development of Internet technology and the widespread use of IoT devices have rendered cyberspace an indispensable component of modern society. However, the openness and interconnectivity of networks also introduce security risks, with network security issues stemming from attacks becoming increasingly frequent, including data leakage and malicious intrusions [1]. As abnormal traffic intrusions pose an escalating threat to network security, the challenge of quickly and accurately detecting such intrusions has emerged as a pressing concern, so the optimization and improvement of network intrusion detection systems has become the focus of attention in the field of network security at present [2]. Currently, deep learning theory [3] has demonstrated remarkable success across various fields, prompting researchers to explore its application in intrusion detection. Deep learning models are adept at learning complex nonlinear features, which enables them to better identify potential threat patterns within network traffic, including anomalous behaviors obscured among substantial volumes of normal traffic [4]. Nevertheless, the application of deep learning methods to intrusion detection is constrained by the issue of dataset imbalance. In recent years, numerous new network traffic datasets have emerged, some of which include the latest attack vectors. However, the sample sizes for certain anomalous traffic types remain insufficient, resulting in significant disparities in the number of samples across different traffic types within the datasets [5]. This imbalance renders deep learning-based intrusion detection models more responsive to traffic types with larger sample sizes, while neglecting attack traffic with fewer samples that are critical to network security, ultimately compromising overall accuracy [6]. Therefore, it is essential to preprocess datasets using relevant balancing techniques, such as oversampling or undersampling, to enhance the accuracy of intrusion detection models [7].

To address these challenges, this paper presents an innovative approach to network intrusion detection that integrates hybrid sampling techniques with deep learning architectures to enhance detection accuracy. The proposed hybrid sampling method employs the Borderline SMOTE algorithm to augment the representation of the minority class, followed by the application of the OSS algorithm to filter out noise points from the majority class and eliminate minority samples that are excessively proximate to the majority class. Concurrently, this approach reduces the sample size of the majority class to achieve a balanced class distribution and mitigate the risk of overfitting. To tackle the complexity inherent in data features, the study further refines the structure of the deep learning model to bolster intrusion detection efficacy, culminating in the development of a network model designated as TRBMA. This model comprises two parallel modules: the first module employs Temporal Convolutional Networks (TCNs) to address the limitations of the one-dimensional ResNet network, particularly its inadequacy in capturing long-range dependencies in time-series data. The second module utilizes a Bidirectional Gated Recurrent Unit (BiGRU) network, which effectively learns the dynamics among neighboring elements in a sequence in both forward and backward directions. Additionally, the two modules are interconnected through a multi-head self-attention mechanism, which facilitates the parallel learning of correlations across various subspaces of the data through multiple distinct “heads”, thereby enhancing the model’s expressive capacity. Consequently, the TRBMA model is capable of comprehensively learning both local features and long- and short-range correlation features of the data, achieving superior feature extraction and fusion via the multi-head self-attention mechanism, ultimately leading to improved detection performance. Finally, the CIC-IDS-2017 dataset is balanced utilizing the BS-OSS hybrid sampling method, which combines OSS undersampling with Borderline SMOTE oversampling, and this balanced dataset is employed for training the TRBMA model, resulting in enhanced model performance and generalization capabilities.

The primary innovations and contributions of this research are outlined as follows:A novel intrusion detection model, designated as TRBMA, has been proposed. This model integrates a one-dimensional Temporal Convolutional Network (TCN) with ResNet architecture and a Bidirectional Gated Recurrent Unit (BiGRU) in a parallel configuration. Additionally, it incorporates a multi-head self-attention mechanism, which enhances its capacity to extract more nuanced and comprehensive features from time-series data, thereby improving classification performance.A novel hybrid sampling technique, referred to as BS-OSS, which integrates the Borderline SMOTE algorithm with the OSS algorithm, has been introduced as an effective approach to address the issue of class imbalance within the dataset. This method aims to enhance both the performance and generalization capabilities of deep learning-based models for network intrusion detection.A comprehensive series of intrusion detection experiments was performed utilizing the publicly available CIC-IDS 2017 dataset. The results of these experiments, which involved comparisons with various models and methodologies, substantiate the superior performance of the intrusion detection model presented in this study. Furthermore, the findings highlight the effectiveness and efficiency of the hybrid BS-OSS sampling technique employed.

## 2. Related Works

Conventional machine learning models often depend on manual feature extraction and exhibit restricted model complexity, which constrains their capacity to manage large-scale datasets and to identify intricate patterns within the data. In contrast, deep learning models demonstrate enhanced representational and generalization abilities, as they can automatically extract features through end-to-end learning processes and effectively capture complex and abstract patterns in the data by employing multilayer architectures [8]. In the existing literature, Wu et al. [9] employed a Deep Belief Network (DBN) for feature extraction, which was subsequently utilized alongside a Support Vector Machine (SVM) classifier. However, the model exhibited inconsistent accuracy across various attack types and demonstrated elevated false alarm rates, indicating a pressing need for an adaptive model capable of accommodating diverse network architectures. He et al. [10] opted for a hybrid approach that combines Pyramid Convolution (PyConv) with Deep Separable Convolution (DSC) to effectively manage multi-scale features while reducing network complexity. Chen et al. [11] proposed the implementation of a fused convolutional neural network (FCNN) to extract multidimensional features from the dataset, addressing the limitations of existing intrusion detection models, particularly the complexity of the feature extraction process and insufficient information retrieval. Zhao et al. [12] proposed a novel Deep Maxout Fusion Convolutional Neural Network (DMFCNN) enabled hybridization model for DDoS attack detection. EL-GHAMRY et al. [13] conducted a comparative analysis of CNN-based models, specifically VGG16, Inception, and Xception, ultimately selecting the most effective CNN model for hyper-parameter optimization to achieve optimal alignment between the dataset and the model. Ren et al. [14] integrated CNN with an attention mechanism to create CA blocks, which significantly enhanced the detection rate of minority classes by stacking multilayered CA blocks to thoroughly learn the multilevel spatio-temporal features inherent in cyber-attack data. In a similar vein, Kamal et al. [15] proposed a deep learning model that integrates Transformer and CNN architectures. The Transformer component is utilized for the extraction of contextual features, thereby enhancing the system’s capability to analyze relationships and patterns within the data effectively. Meanwhile, the CNN component is tasked with the final classification, processing the extracted features to accurately identify specific types of attacks. Their methodology demonstrated remarkable performance, achieving a binary classification accuracy of 99.71% and a multi-class classification accuracy of 99.02% on the NF-UNSW-NB15-v2 dataset. Additionally, for the CICIDS2017 dataset, the model attained a binary classification accuracy of 99.93% and a multi-class classification accuracy of 99.13%. Hu et al. [16] combined the BiGRU network with a self-attention mechanism featuring a gating function to filter out irrelevant noise, resulting in a model architecture that exhibited robust capabilities in temporal information extraction and text sequence feature learning. Ma et al. [17] enhanced the BiGRU by incorporating a residual mechanism, resulting in the ResBiGRU, which mitigated network degradation and improved the model’s capacity to extract traffic features. In a similar vein, Liu et al. [18] enhanced the InceptionNet architecture by incorporating residual and attention modules, resulting in a reduction in the mean squared error (MSE) to 0.04749 and an increase in accuracy to 94.195%. Khan et al. [19] addressed spatial and temporal features in network traffic data through the use of Attention-CNN-BiLSTM and TCN models, respectively, achieving commendable results on the UNSW-NW15 dataset, and the results of the ablation experiments indicated that the incorporation of the TCN module resulted in enhanced overall performance of the model. Kilichev et al. [20] introduced an EVCS intrusion detection model that integrates CNN, LSTM, and GRU, achieving 100% accuracy in binary classification experiments. Bukhari et al. [21] leveraged the spatial data collection capabilities of a stacked convolutional neural network (SCNN) alongside the temporal relationship capture capabilities of BiLSTM to effectively identify complex and previously unknown cyber threats. Rani et al. [22] developed a hybrid deep TCN and GRU network by incorporating dilation, residual connectivity, and GRU layers to maintain the long-term dependencies of inertial signals. Xing et al. [23] employed a hybrid approach that integrates parallel Temporal Convolutional Networks (TCN) and Long Short-Term Memory (LSTM) architectures, utilizing a self-attention mechanism to extract spatio-temporal features within Internet of Vehicles (IoV) traffic. Similarly, Xia et al. [24] implemented a parallelization of ResNet utilizing BiGRU to extract local feature long-range dependencies. This approach enables a more comprehensive capture of significant characteristics associated with network intrusions, thereby enhancing detection performance. Their methodology demonstrated superior accuracy in intrusion detection compared to traditional serial network configurations, as noted in reference [25,26], thereby serving as a valuable point of reference.

The aforementioned literature has integrated various networks by enhancing deep learning models, resulting in a degree of improvement in the accuracy of Intrusion Detection Systems (IDS). However, these approaches exhibit certain limitations in feature learning. Predominantly, they concentrate on singular aspects of feature extraction, which hampers comprehensive feature extraction and increases the risk of overfitting. This issue arises because many machine learning techniques tend to overlook certain categories, leading to suboptimal performance; notably, the performance of these neglected minority categories is often of greater significance. It is evident that while deep learning models possess distinct advantages in the realm of intrusion detection and analysis, there remains a critical need to augment the representation of minority category samples to achieve a balanced dataset. This adjustment is essential for enhancing the model’s capability to recognize features across diverse categories, thereby mitigating misclassifications stemming from category imbalance and ultimately improving overall detection accuracy and robustness. Furthermore, Mozo et al. have investigated the significance of balancing the intrusion detection dataset for the training of IDS models, as documented in their study [27]. They empirically demonstrated that synthetic data can yield effects comparable to those of real data within intrusion detection models without adversely impacting the real data. Nonetheless, to generate a sample distribution that meets statistical significance, a requisite amount of prior distribution is necessary for the creation of traffic samples. In instances where sample sizes are insufficient or exceedingly small, the synthetic data may lack the statistical significance required for rigorous experimental analysis.

The issue of dataset imbalance is currently addressed through two primary approaches: the data level and the algorithmic level. For instance, in the context of class imbalance within Intrusion Detection System (IDS) datasets, Imrana et al. [28] implemented an enhanced focal loss function at the algorithmic level. Volk et al. introduced the Adaptive Cost Sensitive Learning (AdaCSL) algorithm, which aims to mitigate the costs associated with misclassified instances resulting from erroneous predictions [29]. At the data level, Soni et al. [30] employ the SMOTE to mitigate bias toward the majority class, thereby achieving class balance in the WSN-BFSF datasets, which contributes to improved performance in intrusion detection. However, the SMOTE method is susceptible to issues such as the introduction of noise samples, duplication of samples, and the potential loss of boundary samples [31]. To address these challenges, Talukder et al. utilized the SMOTE-Tomek resampling technique, which synthesizes majority class samples while eliminating Tomek Links, thereby reducing noise and alleviating overfitting and underfitting concerns, and attained an accuracy exceeding 99.5% across a diverse array of classifiers [32]. Similarly, Jiang et al. first applied the Outlier Subtraction Sampling (OSS) algorithm to diminish noise within the majority class, subsequently augmenting minority samples through the SMOTE algorithm to achieve data balance [33]. Also, to tackle the issue of the prevalence of noisy samples generated by the SMOTE algorithm, particularly those situated within the vicinity of majority class samples, Hu et al. enhanced the SMOTE algorithm by integrating it with the K-means algorithm [16]. Zhang et al. employed a methodology that involved resampling all class samples to achieve a uniform number of instances through the application of SMOTE oversampling in conjunction with GMM-based clustering undersampling. This approach resulted in a notable enhancement of the detection rate for attack classes [34]. Liu et al. employed ADASYN oversampling techniques to achieve balance within the NSL-KDD dataset. This approach resulted in enhancements in the model’s accuracy, recall, and F1-score by 14.03%, 5.28%, and 10.79%, respectively. However, the augmentation of minority class instances following the dataset balancing led to a 1.67% reduction in the overall accuracy of the model [18]. Cao et al. proposed a hybrid sampling method that integrates the ADASYN and RENN algorithms, which oversamples minority class instances using ADASYN while simultaneously undersampling majority class instances with RENN [35]. Furthermore, Yang et al. introduced the Local Outlier Factor (LOF) algorithm to further address noise issues in the resampled dataset, leading to enhanced model performance [36]. Kamal et al. introduced an innovative resampling methodology that incorporates various data resampling techniques, including Adaptive Synthesis (ADASYN), Synthetic Minority Oversampling Technique (SMOTE), Edited Nearest Neighbors (ENN), and the application of class weights, in order to address the issue of class imbalance [34]. Islam et al. developed the KNNOR method by examining the compactness and spatial distribution of the minority class in relation to other classes, thereby identifying critical and safe regions for augmentation and generating synthetic data points for the minority class [37]. Lastly, Guo et al. proposed an adaptive algorithm that integrates Support Vector Machines (SVM) with Borderline SMOTE oversampling to tackle the challenges associated with unbalanced data classification [38]. Tu et al. incorporated the ENN undersampling technique alongside the Borderline SMOTE, leading to enhanced performance metrics across the overall dataset as well as within each of the minority sample categories. This improvement surpasses the outcomes achieved by employing either the SMOTE or Borderline SMOTE methods in isolation. The efficacy of the ENN method lies in its ability to effectively identify and mitigate the noise samples produced by the oversampling process [31].

Based on the analysis of the above literature, despite notable advancements in deep learning methodologies for network intrusion detection, the field continues to grapple with two primary challenges. Firstly, existing models predominantly exhibit a limitation in their feature extraction capabilities, often focusing solely on either spatial features (e.g., Convolutional Neural Networks) or temporal features (e.g., Long Short-Term Memory networks or Gated Recurrent Units). This unidimensional approach neglects the potential for synergistic learning that integrates localized patterns with long-range temporal dependencies, thereby constraining the models’ effectiveness in recognizing complex attack patterns. Secondly, while various resampling strategies have been employed to address the issue of class imbalance, traditional techniques such as Synthetic Minority Over-sampling Technique (SMOTE) are susceptible to introducing boundary noise. Furthermore, hybrid sampling schemes have not sufficiently optimized the generation of minority class samples while effectively eliminating noise from the majority class. It is important to note that existing research often treats data balancing and model optimization as separate processes, which hinders the establishment of a systematic integration of the two, ultimately limiting the robustness of the models in practical applications.

To tackle these challenges, this paper proposes a data-model co-optimization framework for intrusion detection. In contrast to the sequential feature extraction architectures prevalent in the literature, this study introduces an innovative parallelized architecture combining Temporal Convolutional Networks and ResNet with Bidirectional Gated Recurrent Units (TRBMAs). This dual-channel architecture is designed to capture multi-scale local features through one-dimensional dilated convolution while simultaneously modeling both forward and backward temporal dependencies. Additionally, the integration of a multi-head self-attention mechanism facilitates cross-modal feature fusion, thereby overcoming the constraints associated with traditional single-path feature learning. At the data level, the proposed hybrid BS-OSS sampling method distinguishes itself from existing cascade sampling models by innovatively merging the boundary enhancement mechanism of Borderline SMOTE with a dynamic neighborhood cleaning strategy from OSS. This approach enhances the diversity of minority class samples while systematically eliminating interfering samples from the majority class, resulting in a purer decision boundary. Notably, this study also introduces the decoupled weight decay mechanism of the AdamW optimizer into the domain of intrusion detection for the first time, which significantly enhances the model’s generalization capabilities on balanced datasets through the separation of gradient updating and regularization processes. Experimental results demonstrate that the proposed framework achieves high-precision detection of rare attack classes, such as User to Root (U2R) and Remote to Local (R2L), on the CIC-IDS 2017 dataset. Furthermore, the visual analysis of multi-head attention weights substantiates the model’s proficiency in emphasizing critical temporal features, thereby offering a novel technological approach for intrusion detection in complex network environments.

## 3. Methodology

### 3.1. Borderline SMOTE-OSS Hybrid Sampling Method

The Borderline SMOTE algorithm enhances the distribution of classes within a dataset by generating synthetic samples exclusively in the borderline regions of minority class samples [39]. This approach involves categorizing minority samples into three distinct groups: “Safe”, “Danger”, and “Noise”. The “Safe” category encompasses samples that are predominantly surrounded by other minority samples, while the “Noise” category includes samples that are primarily surrounded by majority samples. The “Danger” category consists of samples that are situated in boundary regions, characterized by being surrounded by more than half of the majority samples. Only the samples classified as “Danger” are subjected to the SMOTE oversampling process. This entails identifying the K nearest neighbors for each “Danger” sample and generating new synthetic samples that lie between these neighbors. These newly created samples are then incorporated into the minority sample set, thereby augmenting the number of minority samples, reinforcing the boundary region, mitigating the issue of class overlap, and enhancing the model’s capacity to accurately identify minority samples. The methodology for generating synthetic samples for boundary samples is illustrated in Figure 1.

One-Sided Selection (OSS) undersampling integrates Tomek links undersampling with KNN undersampling techniques. This method not only eliminates noise and boundary samples from the majority class but also identifies and discards redundant samples, thereby producing a dataset characterized by a more balanced and informative class distribution. The operational procedure of the OSS undersampling method is as follows: it commences by initializing a set C that includes all minority class samples along with one randomly selected majority class sample. Subsequently, a nearest-neighbor classifier is trained using set C to classify the original training dataset S [40], with misclassified samples being incorporated into set C. This iterative process continues until no additional samples are added to set C. Following this, Equation (1) is employed to assess whether pairs of samples that are in close proximity across different classes are each other’s K Nearest Neighbors. If this condition is met, the majority class samples from both classes are removed, thereby aligning the number of majority class samples with that of the minority class samples and achieving class balance [41]. Ultimately, this procedure yields a balanced dataset that effectively eliminates potential noise and reduces the prevalence of majority class samples.(1)fⅆxi,xj=∑k=1nxi,k−xj,k2

Consequently, this study introduces a novel methodology that integrates OSS undersampling with Borderline SMOTE oversampling, referred to as hybrid BS-OSS sampling. This approach employs the Borderline SMOTE algorithm to augment the sample size of the minority class. Simultaneously, the OSS algorithm is utilized to filter the samples, eliminating noise points from both the majority and minority classes, particularly those minority samples that are in close proximity to the majority class. Additionally, this method aims to decrease the number of samples in the majority class, thereby achieving a more balanced class distribution and mitigating the risk of overfitting. The procedural steps of the BS-OSS algorithm are shown in Algorithm 1.
**Algorithm 1** BS-OSS Hybrid Sampling Algorithm**  Input:** Training set S, test set T, number of nearest neighbors K=3, desired balance ratio ratio**  Output:** Balanced dataset Sbalenced
**  Begin**  1.**Initialize** C=∅  2.**While** ∣C∣<ratio×∣S∣ **do**  3.  **For each** minority class sample x∈S **do**  4.    Compute distances to majority class samples and identify borderline samples BS
  5.  **For each** x∈BS **do**                    **For** k=1 **to** K **do**                        Select one nearest neighbor $x_{neighbor}$
                        Generate synthetic samples $S_{new}$ by interpolating between x and $x_{neighbor}$
                        Add $S_{new}$ to C
  6.    **End for**  7.  **End for**  8.  Randomly select majority class samples and add to C until ∣C∣=ratio×∣S∣
  9.**End while**  10.**Train** a 1-NN classifier on C
  11.**Classify** S using the 1-NN classifier  12.**For each** misclassified majority class sample x **do**  13.  Add x to C  14.**End for**  15.**Apply** Tomek Links to C to remove noisy and borderline samples  16.**Output** balanced dataset $S_{balenced}=C$**End**

Through the aforementioned steps, the BS-OSS algorithm addresses the issue of class imbalance in unbalanced datasets comprehensively, while mitigating the risks of overfitting and class overlap, ultimately resulting in a balanced dataset characterized by minimal noise and distinct categorical boundaries.

### 3.2. The Architecture of the TRBMA Model

This study introduces a network intrusion detection model designated as TRBMA, which employs the AdamW optimizer during the training phase. Unlike conventional optimizers, AdamW incorporates weight decay in the parameter update process, thereby facilitating accelerated model convergence and enhancing training efficiency. The architecture of the model comprises two parallel network modules: the 1D-TCN-ResNet module and the BiGRU module.

Initially, the TCN-ResNet module is adept at extracting local features and long-term temporal dependencies from parallel time-series data across various time scales. Temporal Convolutional Networks (TCN) proficiently encapsulate the long-term dependencies present in time-series data by utilizing an expanded causal convolution approach. This is because the dilated convolution in TCN exponentially expands the receptive field. For a kernel size k=3 and dilation rate d=2, the effective receptive field after n layers is:(2)RTCN=k−1⋅Σⅈ=0n−1di+1

This allows TCN to capture long-term dependencies (e.g., multi-step attack sequences). This expansion factor enables TCN to encompass a broader temporal window, thereby addressing the limitations associated with conventional Residual Networks (ResNet) in the context of time-series modeling. Additionally, the residual architecture of ResNet facilitates improved gradient propagation through inter-layer connections, and the integration of these two methodologies allows for a hierarchical extraction of both local and global features. This module features a modified version of ResNet, adapted to a one-dimensional format that is more suitable for time-series analysis. The 1D convolutional layers in ResNet extract fine-grained local patterns (e.g., sudden traffic spikes) through hierarchical stacking:(3)$$F_{ResNet}\left(x\right)={Conv1D}_{k=3}\left(ReLU\left(BN\left({Conv1D}_{k=3}\left(X\right)\right)\right)\right)$$

The combined output $H_{TCN-ResNet}=F_{TCN}\left(x\right)=F_{ResNet}(x)$ ensures both the global context and local detail are preserved.

Concurrently, the BiGRU module employs both forward and backward GRU mechanisms to effectively capture bidirectional dependencies and local patterns within the time-series data, to obtain $h_{t}=GRU(x_{t},h_{t-1})⊕GRU(x_{t},h_{t+1})$, thereby enriching the understanding of the dynamic interactions among neighboring elements in the sequence and elucidating the evolutionary trends of the data [42].

Subsequently, a multi-head self-attention mechanism is integrated following both the TCN-ResNet and BiGRU networks to enhance the modeling of inter-feature relationships.

The MHA mechanism emphasizes significant temporal intervals by utilizing attention weights, such as those defined by the formula $Attention\left(Q,K,V\right)=Softmax\left(\frac{QK^{T}}{\sqrt{d_{k}}}\right)V$, and the integration of these weights facilitates an adaptive fusion of both local and global features [43]. The complementary feature information derived from the parallel 1D-TCN-ResNet and BiGRU modules is then amalgamated [44].

This integration of the 1D-TCN-ResNet and BiGRU modules facilitates a comprehensive extraction of features from time-series data, thereby augmenting the model’s expressive capacity. Ultimately, the final outputs are generated through a fully connected layer.

The structural representation of the model is illustrated in Figure 2.

As illustrated in Figure 1, the model comprises three primary components: the TCN-ResNet module, the BiGRU module, and the feature fusion output module. In the TCN-ResNet module, input data structured as (None, 78, 1) is initially processed by the TCN, where 78 represents the number of feature parameters and the channel count is 1. The TCN is particularly effective for analyzing time-series data, as it captures long-term dependencies by expanding the receptive field through dilation convolution [29]. Consequently, positioning the TCN prior to the ResNet facilitates the early acquisition of critical temporal features by the network. The TCN layer incorporates a dilation convolution layer, followed by a ReLU activation function, batch normalization, and a max pooling layer to mitigate model complexity. The subsequent architecture consists of two residual blocks, each comprising two layers of convolutional networks with a kernel size of 3. The feature maps are resized using a 1 × 1 convolution. The feature matrices from the two branches of the residual blocks are aggregated and subsequently processed through a ReLU activation function. The resulting feature map is then transformed into a one-dimensional feature vector of shape (None, 512) via global average pooling and spreading operations, which is subsequently input into the Multi-Head Attention module. In the BiGRU module, the input data must first be transposed to conform to the input shape requirements of the Gated Recurrent Unit (GRU) [45]. The output from the last time step of each sequence is then extracted as the feature representation using both forward and backward GRUs. Following this, the outputs from the TCN-ResNet and BiGRU modules are directed into the feature fusion output module. This module employs the Multi-Head Attention mechanism to focus on distinct aspects of the inputs across various representation subspaces, thereby capturing more nuanced and comprehensive feature representations. The outputs from the two modules are concatenated to form a unified feature vector, enhancing the model’s comprehension of the time-series data. Ultimately, the model produces final predictions of shape (None, 11) through a fully connected layer.

The overall mathematical principles of the TRBMA model are as follows:$${TRBMA}_{features}=F_{TCN-ResNet}\left(x\right)⊕F_{BiGRU}\left(x\right)$$

In this context, $F_{TCN-ResNet}\left(x\right)$ denotes the local features and long-term dependencies that are derived from the TCN-ResNet module. $F_{BiGRU}\left(x\right)$ signifies the dynamic temporal features obtained from the BiGRU module, both methodologies circumvent the issue of feature confusion commonly associated with traditional tandem models by employing a parallel architecture, while ⊕ represents the weighted fusion achieved through the application of MHA.

The specific theoretical basis of the above modules is described as follows.

### 3.3. Residual Network (ResNet)

The model delineated in this study incorporates an enhanced ResNet network architecture. ResNet, introduced by Kaiming He’s research team in 2016, was developed to address the issues of gradient vanishing and network degradation encountered during the training of deep neural networks [42]. This architecture has gained widespread application in various domains, including image classification, object detection, and semantic segmentation. Unlike conventional convolutional neural networks (CNNs), ResNet incorporates residual connections that allow the input signal to bypass specific layers, thereby facilitating the summation of outputs from these layers [46]. This innovative approach effectively mitigates the challenges associated with gradient vanishing and degradation. The process of residual connectivity is illustrated in Figure 3.

Let X represent the input, $F\left(x\right)$ denote the residual mapping, and $H\left(x\right)$ signify the output. The relationship among these three components can be expressed as follows:(4)$$H\left(x\right)=F\left(x\right)+X$$

The specific architecture of the residual block is illustrated in Figure 4. In contrast to the conventional ResNet architecture, which is typically employed for processing two-dimensional data, the network traffic data utilized in this study is characterized as one-dimensional time-series data. Consequently, all convolutional layers within the residual block of the ResNet module have been adapted to one-dimensional convolutional layers in the proposed model. This modification aims to enhance the model’s capacity to capture temporal or sequential features within the data sequences. Each residual block comprises two convolutional layers, each utilizing a 3 × 1 kernel, thereby maintaining an identical number of output channels. Here, “3” denotes the sequence length, while “1” represents the width of the feature dimension. Each convolutional layer is succeeded by a batch normalization layer and a ReLU activation layer. The two convolution operations are interconnected via a cross-layer data pathway, with the input being directly added prior to the final ReLU activation function. By facilitating connections across different layers, the gradient can be directly transmitted to the shallower layers during the backpropagation process, thereby mitigating the issue of vanishing gradients [42]. In mathematics, the residual gradient formula is as follows:(5)$$\frac{\partial H\left(x\right)}{\partial \chi }=\frac{\partial F\left(x\right)}{\partial \chi }+1$$

In general, even when the $\frac{\partial F\left(x\right)}{\partial \chi }$ approaches zero, the gradient can still be effectively propagated through identity mapping, which involves a simple addition of one. This characteristic notably enhances the training stability of deep neural networks [42], consequently mitigating the likelihood of overfitting that may arise from network degradation.

To accommodate changes in the number of channels, an additional 1 × 1 convolutional layer is incorporated to reshape the input before the summation operation. This architectural design significantly enhances the model’s performance and stability by increasing the number of residual blocks, thereby enabling the network to achieve considerable depth while maintaining manageable computational complexity.

### 3.4. Temporal Convolutional Network (TCN)

The Temporal Convolutional Network (TCN) is employed in the model presented in this study to enhance the architecture of the ResNet network. The TCN was initially introduced by Bai et al. in 2018 as an advanced framework for time-series modeling, building upon the principles of Convolutional Neural Networks (CNNs) [43]. The TCN distinguishes itself through its implementation of dilated causal convolution and residual modules, which enable it to maintain longer temporal dependencies compared to traditional CNNs, thereby enhancing its capacity to effectively capture features within time-series data. The dilated convolution utilized in TCN involves the introduction of “gaps” between the components of the convolutional kernel. This modification serves to enlarge the receptive field of the convolution while maintaining the original dimensions of the kernel, thereby facilitating the effective capture of long-term dependencies within the data. The size of its sensory field can be mathematically expressed by the mathematical Equation (6), where k represents the size of the convolutional kernel and d denotes the dilation factor. This demonstrates that TCN effectively captures long-range temporal dependencies through exponential expansion.(6)$$R=\left(k-1\right)\times d^{layers}+1$$

Assuming a convolution kernel size of k=3 and a dilation rate of d=2, with n=4 and R=31, TCN demonstrates a linearly increasing receptive field in contrast to standard convolutions, which have a receptive field of $R=3^{n}$. This characteristic allows TCNs to achieve logarithmic complexity [43], thereby enhancing their ability to effectively capture long-term patterns in attack traffic, such as the sustained nature of DDoS attacks. Furthermore, this approach mitigates the risk of underfitting that may arise from inadequate local feature learning.

The causal convolution employed in TCN guarantees that the output is dependent solely on present and past inputs, thereby preventing the inadvertent incorporation of future information. This is mathematically represented as follows:(7)F∗ⅆXxt=∑k=0K−1fk⋅Xt−k⋅d

Among them, F is the convolution kernel and d is the expansion factor, ensuring the rigor of temporal modeling.

In contrast to conventional Recurrent Neural Network (RNN) architectures typically employed for sequence data processing [47], the TCN leverages causal convolution to facilitate the modeling of extended time series [48]. This convolutional approach allows for more efficient utilization of Graphics Processing Units (GPUs) for parallel computation. Furthermore, the incorporation of residual connections mitigates issues related to gradient explosion and vanishing gradients commonly encountered in RNNs. Consequently, this paper employs the TCN to augment the ResNet architecture, facilitating the model’s ability to learn temporal features at an earlier stage. The structural design of the TCN is illustrated in Figure 5.

The computation of atrous convolution is expressed as follows:(8)F∗ⅆXxt=∑k=1Kfkxt−K−kⅆ1

In this context, $F=\left(f_{1},f_{2},\cdots ,f_{k}\right)$ represents the convolution kernel, $X=\left(x_{1},x_{2},\cdots ,x_{t}\right)$ denotes the input sequence, ∗ signifies the convolution operator, and ⅆ refers to the expansion coefficient associated with an atrous convolution. Ordinary causal convolution represents a specific instance when *d* = 1, wherein the receptive field of atrous convolution is defined as (k−1)d+1, with k denoting the size of the convolution kernel. The importance of this expansion lies in its capacity to enable the TCN to achieve a broader receptive field while maintaining a more shallow network architecture.

### 3.5. Bidirectional Gated Recurrent Unit (BiGRU)

The model presented in this study incorporates a Bidirectional Gated Recurrent Unit (BiGRU) network architecture. BiGRU is frequently employed in deep learning architectures for the analysis of sequential data due to its proficiency in effectively capturing both backward and forward dependencies within a time series [16]. The configuration of the BiGRU network is illustrated in Figure 6.

In Figure 5, $x_{i}\left(i=1,2,3,\ldots \right)$ denotes the input at the present moment, while $h_{i}\left(i=1,2,3,\ldots \right)$ represents the bidirectional hidden state output, which is the concatenation of the hidden states $\underset{h_{i}}{\to }$ and $\underset{h_{i}}{\leftarrow }$ produced by the forward and backward GRUs at time i, respectively. The forward Gated Recurrent Unit (GRU) is designed to capture historical dependencies, while the backward GRU is responsible for capturing future dependencies. The combined output is represented as $h_{t}=\left[\underset{h_{i}}{\to };\underset{h_{i}}{\leftarrow }\right]$, which integrates both $\underset{h_{i}}{\leftarrow }$ and $\underset{h_{i}}{\to }$, thereby encompassing a more extensive range of contextual information.

The BiGRU enhances the extraction of features from time-series data by concurrently processing the input in both forward and reverse directions. At each time step, the input is fed into two independent GRUs operating in opposite orientations; one GRU analyzes the forward sequence while the other examines the reverse sequence. The final output is generated by combining the outputs of these two unidirectional GRUs. This operational framework allows the BiGRU to capture comprehensive contextual information at each time step, facilitating the learning of temporal relationships between preceding, succeeding, and current states. Consequently, it can uncover underlying temporal patterns within the time-series data, thereby significantly enhancing the learning capacity of the neural network.

In contrast to conventional GRU, which depend exclusively on past data ($\underset{h_{i}}{\to }=GRU(x_{t},h_{t-1})$), BiGRU incorporate both forward causality and backward correlation via bidirectional propagation ($h_{t}=GRU(x_{t},h_{t-1})⊕GRU(x_{t},h_{t+1})$). Consequently, the dimensionality of the final hidden state, denoted as $h_{t}=\left[\underset{h_{i}}{\to };\underset{h_{i}}{\leftarrow }\right]$, is effectively doubled, and the information entropy associated with this state can be expressed as:$$H\left(h_{t}\right)=H\left(\underset{h_{i}}{\to }\right)+H\left(\underset{h_{i}}{\leftarrow }\right)-I(\underset{h_{i}}{\to };\underset{h_{i}}{\leftarrow })$$
where I(⋅) is mutual information.

### 3.6. Multi-Head Attention (MHA)

The model presented in this paper employs a Multi-Head Attention (MHA) mechanism to effectively capture features across various scales and regions, thereby improving the model’s capacity to comprehend intricate scenes. The MHA mechanism serves as a fundamental element within the Transformer architecture, significantly augmenting the model’s expressive capacity. This is achieved by executing multiple Self-Attention layers concurrently and integrating their outputs, thereby enabling the model to simultaneously capture information from the input sequence across various subspaces [42]. In the MHA mechanism, the initial step involves processing the input sequence through Equations (9)–(12) to derive the Query, Key, and Value representations via three distinct linear transformation layers. Subsequently, these transformed vectors are partitioned into multiple “heads”, each possessing its own independent matrices for Query, Key, and Value. For each head, a self-attention operation is executed, as delineated in the relevant equations. Ultimately, the outputs from all heads are concatenated and subsequently integrated through a linear layer to produce the final attention output vector.(9)$$Query=W^{q}\cdot X$$(10)$$Key=W^{k}\cdot X$$(11)$$Value=W^{v}\cdot X$$(12)$$Self-Attention\left(X\right)=Softmax\left(\frac{{XW}^{Q}\cdot {{(XW}^{K})}^{T}}{\sqrt{d_{k}}}\right)\cdot {XW}^{V}$$

In this context, let *X* denote the input feature sequence, while $W^{q}$, $W^{k}$, and $W^{v}$ represent the trainable matrices that serve as parameters to be optimized. The symbols *Q*, *K*, and *V* correspond to Query, Key, and Value, respectively. Additionally, $d_{k}$ signifies the scaling factor, which is equivalent to the dimension of the input sequence. The scaling operation is executed by dividing the result of the dot product by $d_{k}$, thereby effectively regulating the variance from $d_{k}$ to 1, which helps mitigate the issue of vanishing gradients. The operational flow of the MHA mechanism is illustrated in Figure 7.

## 4. Experiments

### 4.1. Dataset Description

The dataset selected for analysis in this study is the CIC-IDS-2017 network traffic intrusion detection dataset, which has been developed through a collaborative initiative between the Communications Security Establishment and the Canadian Institute for Cybersecurity. This dataset is utilized to assess the efficacy of the methodology proposed herein. It encompasses 14 distinct categories of attack traffic alongside one category of normal traffic, comprising a total of 78 feature fields and 15 category labels. The various sub-datasets within the CIC-IDS-2017 dataset, along with the distribution of the associated traffic categories, are presented in Table 1.

### 4.2. Dataset Preprocessing

The data preprocessing in this chapter’s experiments includes four steps: data cleaning, data numericalization, data normalization, and BS-OSS Hybrid Sampling.

#### 4.2.1. Data Cleaning

Initially, 1362 rows of sample data containing “NULL” and “NaN” values were eliminated from the dataset. Subsequently, the disproportionately represented network traffic attack categories, namely, “Infiltration” and “Heartbleed,” were also removed. The respective counts of traffic data for these categories were 36 and 11, accounting for merely 0.002% of the total dataset, thereby exerting minimal influence on the overall analysis. Finally, similar attack types, specifically Web Attack-Brute Force, Web Attack-SQL Injection, and Web Attack-XSS, which exhibited low prevalence, were consolidated into a single category labeled “Web Attacks”. This consolidation aimed to reduce the number of categories and enhance the model’s efficiency. The distribution of traffic across each category in the CIC-IDS-2017 dataset, following the data cleaning and integration process, is presented in Table 2.

#### 4.2.2. Data Numericalization

The experiments presented in this chapter involve the transformation of character-type data in the label column of the original CIC-IDS-2017 dataset into numeric values ranging from 0 to 10. This conversion is conducted in accordance with the specific categories of the sample data.

#### 4.2.3. Data Normalization

The experiments presented in this study employ the Min–Max normalization technique to facilitate data normalization. Specifically, continuous data are standardized to a range of [0, 1] through the application of the Min-Max normalization method. This approach enables the model to more effectively manage the distribution of diverse data and mitigates the impact of outliers. Subsequently, the pre-processed dataset is divided into training and testing subsets in an 80:20 ratio.

#### 4.2.4. BS-OSS Hybrid Sampling

The Borderline SMOTE algorithm is employed to oversample the attack traffic samples associated with the DDoS, DoS GoldenEye, DoS slowloris, DoS Slowhttptest, FTP-Patator, SSH-Patator, Bot, and Web Attack categories, which represent seven classes of attack traffic characterized by a limited number of samples. This approach aims to enhance the representation of these classes. Subsequently, all classes of samples undergo OSS undersampling for the purposes of sample screening and denoising, which concurrently reduces the number of samples in the majority of classes. Ultimately, the ratio of normal traffic samples to abnormal attack traffic samples in the binary training set, following the implementation of BS-OSS hybrid sampling, is approximately 1:1. Similarly, the ratio of the various categories of network traffic samples in the multi-classification training set is also approximated to be 1:1. Table 3 presents the quantity of samples in the dataset prior to and subsequent to the implementation of hybrid sampling.

### 4.3. Experimental Environment

This paper primarily utilizes the Python programming language and the PyTorch 1.10.2 framework for model construction. The specific experimental environment configuration is shown in Table 4.

### 4.4. Model Parameter Design

To assess the efficacy of the intrusion detection model, a comparative analysis was conducted between the proposed model presented in this chapter and existing methodologies documented in the literature. The finalized configurations of the hyper-parameters utilized in the experiments are detailed in Table 5.

### 4.5. Evaluation Metrics

In order to comprehensively assess the detection performance of the model, this study employs accuracy, precision, recall, F1-score, and ROC curve as evaluation metrics, derived from the multi-classification confusion matrix [49]. The confusion matrix is presented in Table 6.

Accuracy represents the ratio of the total number of correctly classified instances to the total number of instances, as shown in Equation (13):(13)$$Accuracy=\frac{TP+TN}{TP+TN+FP+FN}$$

Precision represents the proportion of true-positive samples among those classified as positive samples, as shown in Equation (14):(14)$$Precision=\frac{TP}{TP+FP}$$

Recall represents the probability that a classifier correctly identifies positive instances, as shown in Equation (15):(15)$$Recall=\frac{TP}{TP+FN}$$

F1−score is a harmonic mean of recall and precision, as shown in Equation (16):(16)$$F1-Score=2\times \frac{Precision\cdot Recall}{Precision+Recall}$$

Among them, TP denotes the number of true positives, FP denotes the number of false positives, TN denotes the number of true negatives, and FN denotes the number of false negatives.

The Receiver Operating Characteristic (ROC) curve is a widely utilized instrument for assessing the efficacy of binary classification models. It illustrates the variation in model performance across different threshold values. The horizontal axis of the ROC curve represents the FPR, while the vertical axis denotes the TPR. The mathematical expressions for TPR and FPR are delineated in Equations (17) and (18), respectively. Generally, a ROC curve that approaches the upper left corner signifies superior classifier performance, with the corresponding threshold being regarded as the optimal threshold. Furthermore, a smooth and convex ROC curve typically suggests that the classifier exhibits high classification accuracy and is free from overfitting.(17)$$TPR=\frac{TP}{TP+FN}$$(18)$$FPR=\frac{FP}{FP+TN}$$

## 5. Results and Discussion

### 5.1. Comparative Analysis of Models

To assess the efficacy of the TRBMA model introduced in this chapter in relation to other integrated models, including CNN, TCN, ResNet, BiGRU, CNN-BiGRU, TCN-BiGRU, and TCN-ResNet-BiGRU, comparative experiments were conducted utilizing balanced datasets. These integrated models share structural similarities with the TRBMA model. In the process of substituting or omitting the BiGRU and multi-head self-attention layers from the models presented in this chapter, the parameters of the alternative models were maintained consistently, encompassing aspects such as activation functions, the number of training epochs, and learning rates.

Finally, this study conducts a comparative analysis of the proposed TRBMA model against established intrusion detection techniques, utilizing the CICIDS2017 dataset for evaluation. The performance of the model is assessed through various metrics, including accuracy, precision, recall, and F1-score.

To identify an appropriate learning rate that enhances the model training process and optimizes model performance, this study evaluates four distinct learning rate values: 0.1, 0.01, 0.001, and 0.0001. This approach facilitates a series of controlled experiments aimed at hyper-parameter optimization. As illustrated in Figure 8, the TRBMA model achieves optimal performance at a learning rate of 0.001.

Figure 9 illustrates the variation in accuracy and loss values across iterations throughout the training phase of the TRBMA model. The data indicate that both training and validation accuracies are notably high and stable, while the loss values exhibit a rapid decline followed by a tendency to stabilize. This pattern suggests that the training process of the model has been successful and demonstrates considerable statistical robustness. Furthermore, the performance metrics for both the training and validation datasets are aligned, showing no indications of overfitting or underfitting, thereby affirming the model’s robust generalization capabilities.

After formal training, the performance of each model is shown in Table 7. The designation “N/A” in the table signifies that the information was not referenced in the manuscript.

As illustrated in Table 7, the TRBMA model introduced in this thesis exhibits considerable advantages across several critical performance metrics, achieving an accuracy of 98.66% and an F1-score of 98.67%. The enhancement in performance is attributed to the deliberate optimization of the model architecture and the innovative design of the feature fusion mechanism. A comparative analysis with baseline models reveals that individual architectures, such as CNN, TCN, ResNet, and BiGRU, typically yield accuracy rates below 97%.

The performance gains observed in the combined model substantiate the efficacy of multimodal feature fusion. Specifically, the tandem architecture of TCN-ResNet results in a 2.37% increase in accuracy relative to ResNet alone, thereby validating the strategic placement of the TCN at the front end of ResNet.

The superior accuracy of TCN-ResNet (97.69%) in comparison to ResNet (95.32%) can be ascribed to the enhanced temporal sensory field facilitated by the dilation convolution employed in the TCN, as delineated in Equation (2). The TCN effectively captures temporal dependencies through dilated causal convolution, allowing it to address the limitations of traditional ResNet in time-series analysis by covering a broader temporal window. Concurrently, the residual connection of ResNet, as described in Equation (4), facilitates enhanced gradient propagation via inter-layer connections. This mechanism contributes to the stabilization of gradients and mitigates the degradation commonly associated with deep networks, as evidenced by the rapid convergence of training loss illustrated in Figure 9.

Furthermore, the incorporation of the BiGRU facilitates a comprehensive exploration of temporal features. In contrast to the GRU, which solely depends on historical data ($h_{t}=GRU(x_{t},h_{t-1})$), the BiGRU effectively mitigates the temporal bias associated with unidirectional models. The BiGRU integrates both optimized historical and future contexts by identifying the preceding and subsequent characteristics of attack traffic, such as the initiation and persistence phases of DDoS attacks, through the implementation of bidirectional gating (refer to $h_{t}=GRU(x_{t},h_{t-1})⊕GRU(x_{t},h_{t+1})$). Comparative results indicate that the accuracy of the standalone BiGRU model is 96.64%, with precision at 96.87%, recall at 96.64%, and an F1-score of 96.45%. In contrast, the integration of the TCN-ResNet architecture enhances these metrics to 98.17%, 98.19%, 98.13%, and 98.18%, respectively, thereby confirming the effectiveness of BiGRU networks in extracting temporal patterns. In a comparative analysis of TCN-ResNet (Accuracy = 97.69%, Precision = 97.71%, Recall = 97.68%, F1 = 97.40%) and TCN-ResNet-BiGRU (Accuracy = 98.17%, Precision = 98.19%, Recall = 98.13%, F1 = 98.18%), it is evident that the absence of the BiGRU component in the former model hinders its ability to capture reverse temporal dependencies, such as the delayed responses associated with Heartbleed attacks, thereby resulting in the loss of critical discriminative information.

In the conducted ablation experiments, the removal of the MHA mechanism from the TCN-ResNet-BiGRU model resulted in a 0.51% reduction in the F1-score, thereby underscoring the importance of integrating multi-head attention features. Determine the attention weighting equation utilizing the MHA mechanism outlined below:$$Attention\left(Q,K,V\right)=Softmax\left(\frac{QK^{T}}{\sqrt{d_{k}}}\right)V$$

This allows the model to prioritize critical subspaces (e.g., sudden traffic spikes indicating DDoS attacks). Without MHA, the concatenated features from TCN-ResNet and BiGRU are fused statically, leading to suboptimal integration of local and global patterns.

The TRBMA model employs a parallel multi-head self-attention mechanism, which facilitates a nuanced partitioning of the feature space. Specifically, the TCN-ResNet pathway emphasizes the integration of local convolutional features with global residual features, while the BiGRU pathway concentrates on capturing both forward and backward semantic relationships within sequences. The multi-head attention mechanism effectively decouples these two pathways, allowing for the enhancement of key features across the eight attention heads in a multi-dimensional space. This process culminates in the generation of a more discriminative feature representation through weighted fusion. The dual-channel architecture of the model addresses the traditional residual network’s limitations in capturing long-range dependencies by leveraging TCN’s temporal modeling, while simultaneously preserving the robust feature extraction capabilities of ResNet. Additionally, the parallel BiGRU component facilitates the learning of forward and backward sequence features, resulting in performance improvements exceeding 10% compared to conventional convolutional neural network models. This outcome substantiates the scientific rationale and efficacy of the TRBMA architecture and its integration strategy.

Figure 10 presents the multi-class confusion matrix derived from the predictive classification of the test set following the formal training of the TRBMA intrusion detection model introduced in this chapter. This matrix serves to provide a clearer understanding of the model’s detection performance. The horizontal axis represents the predicted categories, while the vertical axis denotes the actual categories of the data. As illustrated in Figure 9, the classification accuracy for the BENIGN category reaches 98%, demonstrating the model’s effectiveness in distinguishing between normal and attack traffic. Furthermore, the model exhibits an accuracy exceeding 90% for the identification of other traffic types, with the identification accuracies for DoS GoldenEye and SSH-Patator reaching 97% and 96%, respectively. This indicates a high level of accuracy in recognizing these two specific types of anomalous traffic. Conversely, the recognition accuracy for DoS Slowhttptest, Web Attacks, and Bot is comparatively lower at 92%. This reduced accuracy may be attributed to the limited number of samples for these three categories of anomalous traffic within the training set, which may not have been adequately represented for effective learning.

Furthermore, five models were chosen for comparative analysis in this experiment: TGA [48], PSO-GA-ResNet-BiGRU [24], TBGD [25], DMFCNN [13], and PGDOFLN [50]. In a comparative analysis of the TGA and PSO-GA-ResNet-BiGRU models, both of which exhibit a parallel structure, the TRBMA model introduced in this dissertation demonstrates a marked superiority over the TGA. Although the PSO-GA-ResNet-BiGRU model achieves a higher accuracy rate of 99.21%, it exhibits a lower precision of 97.85% and a relatively inadequate recall of 98.17%. The TBGD model, while presenting commendable accuracy (99.08%) and an F1-score of 0.99, shows deficiencies in accurately identifying certain classes within the dataset; specifically, its detection rates for Web_Attack_Sql_Injection and Web_Attack_XSS are alarmingly low at 3% and 6%, respectively. In contrast, the DMFCNN model exhibits a recall rate of 98.75%, whereas the PGDOFLN model achieves a precision rate of 100%. Nonetheless, the PGDOFLN model demonstrates a notable disparity between its recall (95%) and accuracy (95%), suggesting a potential risk of overfitting within its classification approach. In comparison, the TRBMA model introduced in this thesis surpasses these models by approximately 3% in overall detection accuracy for anomalous traffic.

In conclusion, the experimental findings indicate that the TRBMA model demonstrates enhanced performance in managing complex time-series data associated with various types of cyberattack traffic, surpassing that of the individual the TCN network model, ResNet network model, the BiGRU network model, and the TCN-ResNet-BIGRU integrated model. This improvement is particularly notable when the data exhibit significant temporal and spatial characteristics. The TRBMA model not only achieves superior classification outcomes and higher detection accuracy but also exhibits enhanced generalization capabilities. And better than existing models. Furthermore, the comparative argumentation experiments validate the effectiveness, rationality, and stability of the TRBMA model.

### 5.2. Experimental Validation of the TRBMA Model with Integrated BS-OSS

The comparative experiments outlined in this Section are primarily categorized into two distinct groups. The first group aims to assess the efficacy of the TRBMA model in relation to the TRBMA (BS-OSS) model for multi-class classification utilizing the CIC-IDS-2017 dataset. This comparison serves to validate the effectiveness of the TRBMA (BS-OSS) intrusion detection model introduced in this study. The second group focuses on evaluating the accuracy differences between the BS-OSS hybrid sampling method proposed herein and the widely utilized hybrid sampling techniques, in both binary and multi-class classification contexts using the TRBMA model. Throughout both sets of experiments, parameters such as optimizer, batch size, training epochs, learning rate, and weights are maintained consistently, with the experimental environment and evaluation metrics aligning with those established in Section 4.

Figure 11 illustrates the confusion matrix associated with the TRBMA (BS-OSS) model following multi-class classification on the CIC-IDS-2017 test set. A comparison with Figure 9 reveals that the misclassification rate for the TRBMA (BS-OSS) model is lower than that of the TRBMA model, indicating an enhancement in recognition accuracy across all traffic categories. Specifically, the TRBMA (BS-OSS) model demonstrates an increase in recognition accuracy for seven traffic categories—BENIGN, DoS Hulk, PortScan, FTP-Patator, DoS Slowhttptest, Web Attacks, and Bot—enhanced to 99%. And the recognition rate for both DoS GoldenEye and SSH-Patator attack types has reached 100%. Furthermore, it significantly enhances the recognition accuracy for traffic categories that originally contained very few samples in the dataset. Consequently, the TRBMA intrusion detection model, optimized for the CIC-IDS-2017 dataset through the application of BS-OSS hybrid sampling, effectively achieves improved accuracy in identifying both majority and minority categories of anomalous traffic.

Figure 12, Figure 13 and Figure 14 present a visual comparison of the precision, recall, and F1-score performance metrics for the TRBMA model and the TRBMA (BS-OSS) model, utilizing bar charts to illustrate the results on the multicategorical test set. The data depicted in these figures indicate that the TRBMA (BS-OSS) model demonstrates a substantial enhancement in precision, recall, and F1-score across all categories of anomalous traffic.

Figure 12 presents a comparative analysis of the precision metrics for the TRBMA model and the TRBMA (BS-OSS) model across various traffic types within the multi-classification dataset. The results indicate that the precision of the TRBMA (BS-OSS) model, which was trained on a dataset balanced through the BS-OSS technique, demonstrates a notable enhancement in performance for each traffic category. This improvement suggests that the TRBMA (BS-OSS) model exhibits superior capabilities in classifying and identifying malicious intrusion traffic, particularly in scenarios where the sample size is limited.

Figure 13 presents a comparison of the recall values between the TRBMA model and the TRBMA (BS-OSS) model concerning network traffic within a multicategorical dataset. The results indicate a substantial enhancement in the recall values of the TRBMA (BS-OSS) model across all categories of anomalous traffic. This improvement suggests that the balanced dataset, achieved through the application of the BS-OSS technique, facilitated a more comprehensive and effective training process for the model.

Figure 14 presents a comparative analysis of the F1-score between the TRBMA model and the TRBMA (BS-OSS) model across different categories of traffic within the multicategorical dataset. The results indicate that the TRBMA (BS-OSS) model markedly enhances the F1-score for various forms of anomalous traffic. This finding underscores the superior generalization capabilities and robust performance of the TRBMA (BS-OSS) model.

Table 8 presents the impact of various sampling techniques on achieving dataset balance. A comparative analysis of the data in the accompanying table reveals that the BS-OSS method, as proposed in this study, exhibits considerable advantages in addressing unbalanced datasets, applicable to both binary and multi-class classification tasks. Specifically, in the binary classification task, the BS-OSS method achieves an accuracy of 99.43%, while in the multi-class classification task, it attains an impressive accuracy of 99.88%. These results significantly surpass those of other oversampling techniques, including SMOTE, ADASYN, Borderline SMOTE, SMOTE-Tomek Link, and SMOTE-ENN. This evidence suggests that the BS-OSS method is more effective in mitigating data imbalance issues and enhancing the classification performance of the model. Furthermore, the tabulated data indicate that the exclusive application of a singular oversampling method (e.g., SMOTE or ADASYN) may prove inadequate and, in certain instances, detrimental to model performance. For instance, in the multi-class classification task, ADASYN achieves an accuracy of only 97.93%, which is inferior to the 98.25% accuracy of SMOTE. This observation implies that a singular oversampling approach may not sufficiently accommodate the complexities of data distributions and classification challenges. In contrast, the BS-OSS method enhances the delineation of category boundaries by integrating multiple strategies, thereby facilitating a more effective response to these challenges and yielding superior classification outcomes.

Figure 15 illustrates the comparative analysis of the Receiver Operating Characteristic (ROC) curves for the TRBMA model and the TRBMA (BS-OSS) model following binary classification on the CIC-IDS-2017 test dataset. The ROC curve comparison presented in Figure 15 indicates that the ROC curve for the TRBMA (BS-OSS) model is situated closer to the upper left corner, signifying superior performance relative to the TRBMA model. Furthermore, the ROC curve of the TRBMA (BS-OSS) model exhibits a smoother trajectory, suggesting an enhanced classification accuracy.

In order to demonstrate the efficacy of the TRBMA (BS-OSS) model in the context of classification within the field of intrusion detection, this study has chosen sixteen contemporary methodologies from the recent literature for comparative analysis, utilizing the CICIDS2017 dataset as a basis. The results of this comparative analysis are presented in Table 9. The designation “N/A” in the table signifies that the information was not referenced in the manuscript.

A comparative analysis of the data presented in Table 9 reveals that the TRBMA model proposed in this study, in conjunction with the BS-OSS sampling method, demonstrates considerable advantages across multiple metrics. Notably, the proposed method attains accuracy, precision, recall, and F1-score values of 99.88%, 99.86%, 99.88%, and 99.89%, respectively, surpassing the performance of various models and sampling techniques reported in the existing literature.

For example, in the literature [45], GAN are employed to generate a new set of samples in order to mitigate the impact of data imbalance. This approach ultimately yields an impressive 99.86% accuracy, 99.57% precision, 99.74% F1-score, and an exceptional 99.91% recall. Furthermore, it achieves a perfect accuracy of 100% in the identification of normal traffic classified as BENIGN. However, the inherent instability associated with GAN training hampers the generator’s ability to effectively learn the feature distribution of minority samples. Consequently, the quality of the generated minority samples is suboptimal, resulting in the classifier’s accuracy for recognizing minority samples, such as Infiltration and Heartbleed, falling below 95%. This observation is also applicable to the existing literature [51]. Although the studies referenced in [26,52] have made advancements in GAN, the overall accuracy of detection has not seen significant enhancement and continues to be hindered by the persistent issue of low accuracy in identifying samples from a limited number of classes.

The studies referenced in the literature [44,45,46,54] employ a specific oversampling technique, namely, SMOTE or ADASYN or KNN, to augment the samples of the minority class, thereby addressing the issue of data imbalance. Notably, in the literature [49] and literature [18], feature selection methodologies, such as Recursive Feature Elimination (RFE) and Principal Component Analysis (PCA), are incorporated to diminish the dimensionality of the dataset while preserving essential features. However, the reliance on a singular oversampling method in these studies may result in overfitting or a lack of diversity among the generated samples. Furthermore, the relatively simplistic model architectures utilized may not sufficiently capture the intricate characteristics of the data. Consequently, while these approaches can achieve an accuracy of up to 99.94% in binary classification tasks, their performance in multi-class classification is less impressive, with an accuracy of only 96% in detecting anomalous traffic—approximately 3% lower than the model presented in this paper. Nevertheless, the application of RFE and PCA not only enhances detection efficacy but also eliminates less relevant attributes, thereby reducing data dimensionality and significantly decreasing model training time, which helps conserve resources. This outcome also establishes a benchmark for the subsequent research endeavors outlined in this paper.

In comparison to the studies referenced as [16,34], which also employed a hybrid sampling approach, it is noteworthy that both utilized KSMOTE and SMOTE for oversampling in conjunction with Random and GMM for undersampling to achieve dataset balance. Despite the reduced training time associated with their relatively straightforward model structures, these models exhibit limitations in effectively capturing intricate data patterns and features. This challenge is particularly pronounced in high-dimensional intrusion detection datasets that have undergone mixed sampling. Consequently, the performance metrics—accuracy, precision, recall, and F1-score—reported in the literature [16] are approximately 1.5% lower than those achieved by the TRBMA (BS-OSS) method proposed in this thesis. The situation is analogous for the Transformer-CNN, which is a streamlined model that employs a combination of ADASYN, SMOTE, and ENN sampling techniques [15].

In the literature [29,41] and the TRBMA (BS-OSS) model in this paper, the data were balanced with mixed sampling, the difference being that the former two both use tandem spatiotemporal feature extraction and the inputs of GRU and Bi-LSTM are spatial feature vectors extracted by the CNN, which are re-extracted from the CNN-extracted features, instead of using the original traffic information. While in this paper, the parallel spatiotemporal feature extraction method is used in feature extraction, and the original data are used as the input of the temporal and spatial feature extraction module, which is more comprehensive for the feature extraction of the original data.

In comparison to recent high-impact literature [55,56], our proposed method demonstrates an accuracy of 99.88%, surpassing the performance of the Hybrid Learning Model (HLM), which achieves an accuracy of 99.63% [54], and the integrated learning model utilizing graph neural networks, FTG-Net-E, which attains an accuracy of 99.67% [55]. However, it is important to note that our method does not match the accuracy of the S2CGAN model, which incorporates a Siamese Self-Encoder Network (SAN) and a Generative Adversarial Network (GAN) [56] in its ability to identify minority class samples such as Infiltration and Heartbleed. Notably, the S2CGAN model exhibits a 30% lower accuracy in detecting Bot attack traffic compared to our proposed method, TRBMA (BS-OSS), which maintains an accuracy of 99.86% and a recall of 99.88%. This performance indicates that our hybrid architecture effectively addresses the prevalent accuracy–recall tradeoff encountered in classification tasks.

In summary, the experiments outlined in this study demonstrate that the TRBMA (BS-OSS) model exhibits effective detection capabilities for network attacks. Furthermore, it demonstrates robust generalization performance in the face of previously unencountered attacks, and it maintains strong efficacy when applied to datasets characterized by high dimensionality, complexity, and imbalance.

## 6. Conclusions

In this study, we present a novel approach to network traffic intrusion detection, particularly addressing the challenges associated with unbalanced datasets. This method is referred to as TRBMA (BS-OSS), which integrates a temporal convolutional network (TCN), one-dimensional ResNet (1D-ResNet), bidirectional gated recurrent unit (BiGRU), and multi-head attention mechanisms, alongside a hybrid sampling technique known as Borderline SMOTE-OSS. This approach addresses the challenges associated with the inadequate learning of temporal features prevalent in existing network intrusion detection models, which often results in high accuracy yet suffers from significant class imbalance within intrusion detection datasets. Consequently, this imbalance leads to a diminished recognition rate for malicious traffic types characterized by small sample sizes, thereby impairing overall intrusion detection performance. The experimental findings indicate that the TRBMA model achieves a detection accuracy of 98.66% on the CIC-IDS-2017 dataset, with notable enhancements in overall precision, recall, and F1-score when compared to existing models. Furthermore, the dataset processed using the BS-OSS hybrid sampling technique facilitates the training of our model, yielding an impressive detection accuracy of 99.88%. The model’s overall precision, recall, and F1-score are recorded at 99.86%, 99.88%, and 99.89%, respectively, demonstrating a significant improvement over the current state of popular hybrid sampling methods. Verified the rationality and effectiveness of our proposed method. These notable findings highlight the significance of proficient data preparation and the application of advanced deep learning techniques in the context of intrusion detection within the Internet of Things (IoT).

The investigation presented in this thesis highlights the potential benefits of integrating deep learning models to bolster the security of network and cloud environments in the face of progressively advanced cyber threats. Our methodology not only improves real-time detection capabilities but also markedly enhances the performance and generalization of the models by adeptly tackling the issue of data imbalance through the implementation of a hybrid BS-OSS sampling technique, which is a prevalent challenge in the development of IDS.

While this paper has demonstrated commendable outcomes based on the afore-mentioned data, there remain certain limitations that warrant further enhancement in future research endeavors:

In the context of data preprocessing, it is advisable to explore more sophisticated algorithms. For instance, an enhanced version of the Generative Adversarial Network (GAN) may be employed during the data oversampling phase, while Recursive Feature Elimination (RFE) techniques could be utilized for data dimensionality reduction. These approaches aim to enhance the efficacy of preprocessing while simultaneously mitigating computational complexity and memory usage.

The datasets utilized in this study are derived from pre-existing collections, which have been processed by researchers from the original traffic packets. Future investigations should prioritize the analysis of original traffic packets to enable the system to directly process authentic network traffic, thereby ensuring that the research model is applicable in real-world network environments. Additionally, subsequent research should be conducted using a broader array of datasets, such as NSL-KDD, UNSW-NB15, CSE-CIC-IDS2018, and IoT23, to assess the model’s effectiveness, robustness, and generalizability in addressing emerging attack vectors.

The parallel architecture of TRBMA introduces higher computational complexity compared to lightweight models (e.g., single-path CNNs). This may limit its applicability in resource-constrained IoT edge devices. Potential solutions include model pruning [44] or hardware acceleration (e.g., FPGA-based implementations).

Despite these limitations, our framework provides a foundational architecture for hybrid spatiotemporal feature learning in intrusion detection. By openly addressing these challenges, we aim to guide future research toward adaptive, resource-efficient, and attack-resilient IDS designs for next-generation networks. In forthcoming research, we will expand upon the concept of multimodal feature fusion within this parallel heterogeneous model by transforming the traffic data, post-feature dimensionality reduction, into the format of feature maps. This approach aims to uncover latent temporal relationships within raw traffic data. Ultimately, the integration of one-dimensional traffic features with two-dimensional images will synthesize information across modalities, thereby providing a more comprehensive characterization of the data. This methodology is expected to aid in the identification of potential attack vectors and anomalous behaviors, enabling us to adapt to the dynamic nature of network environments and enhance overall network security.

## Figures and Tables

**Figure 1 sensors-25-01578-f001:**
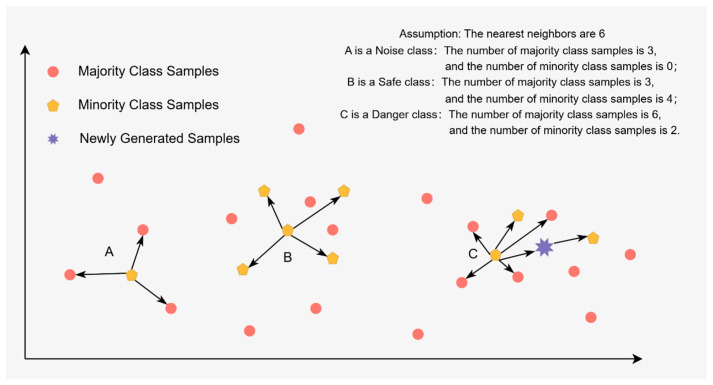
Borderline SMOTE Algorithm Working Diagram.

**Figure 2 sensors-25-01578-f002:**
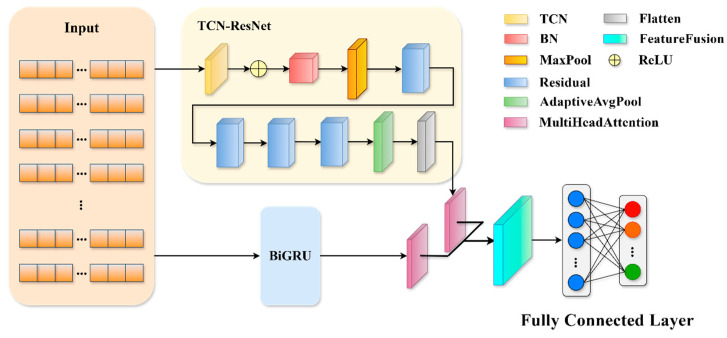
Structure diagram of TRBMA model.

**Figure 3 sensors-25-01578-f003:**
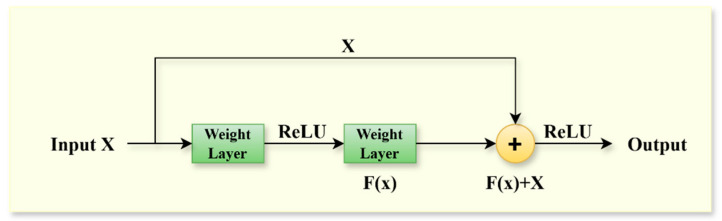
Residual Connection Process Diagram.

**Figure 4 sensors-25-01578-f004:**
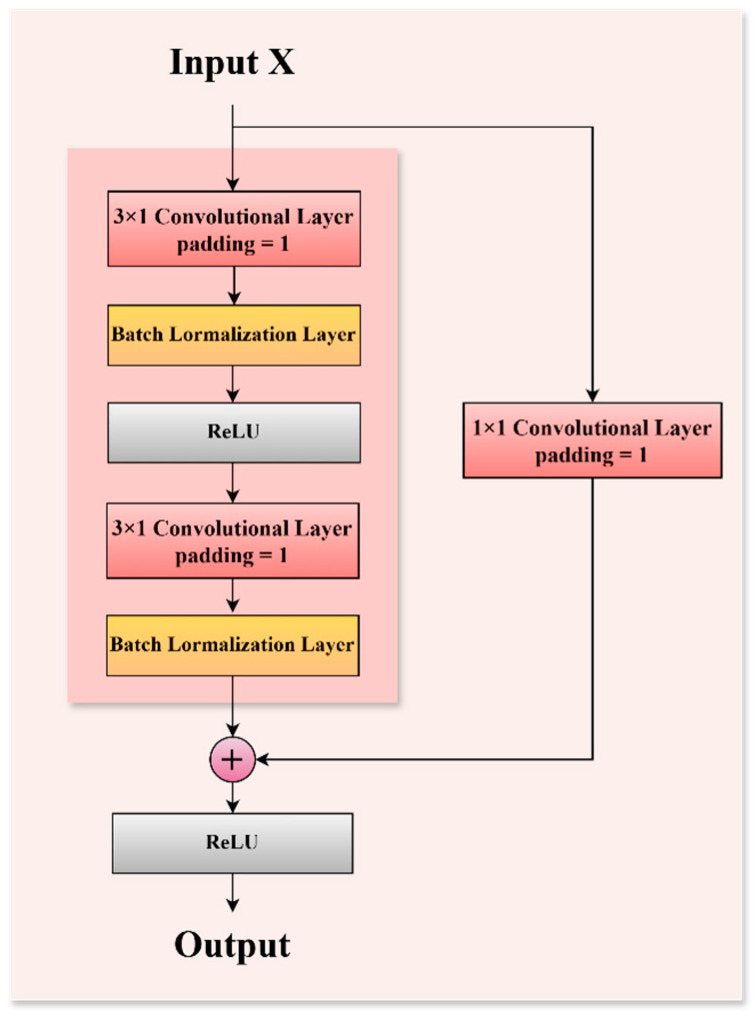
Residual block structure diagram.

**Figure 5 sensors-25-01578-f005:**
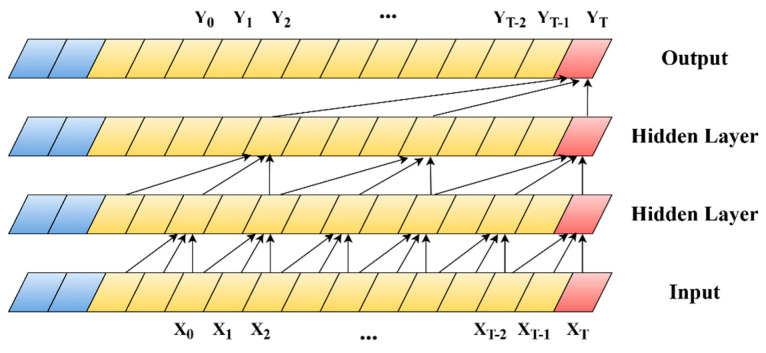
TCN Network Architecture Diagram.

**Figure 6 sensors-25-01578-f006:**
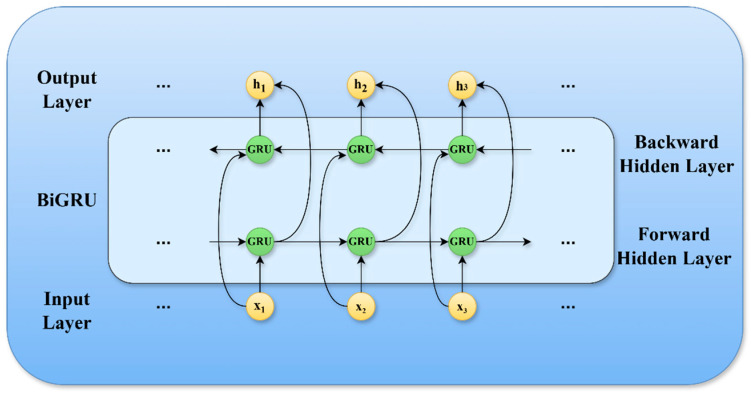
BiGRU Network Architecture Diagram.

**Figure 7 sensors-25-01578-f007:**
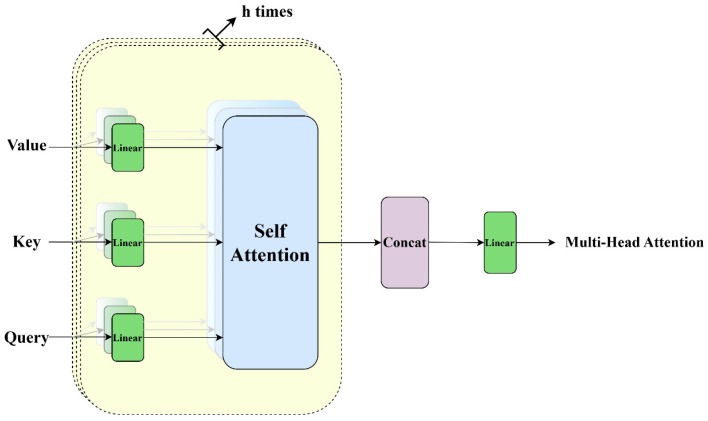
Multi-Head Attention Mechanism Principle Flowchart.

**Figure 8 sensors-25-01578-f008:**
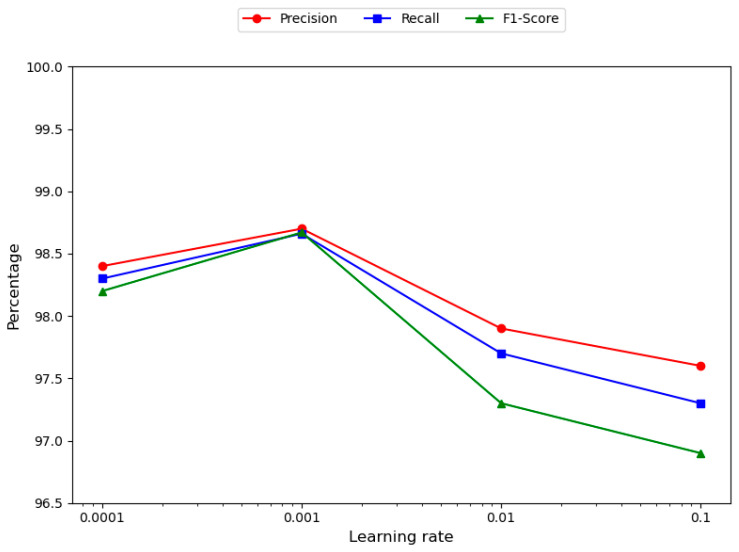
Changes in Precision, Recall, and F1-Score under different learning rates.

**Figure 9 sensors-25-01578-f009:**
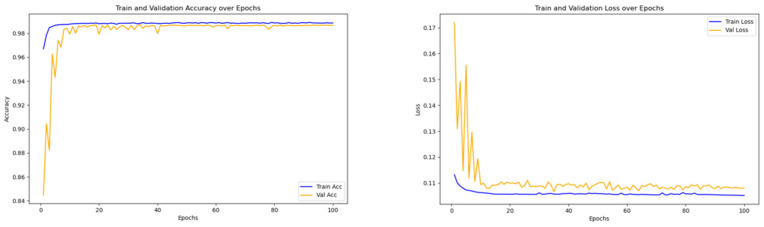
Curves of accuracy and loss values with iterations.

**Figure 10 sensors-25-01578-f010:**
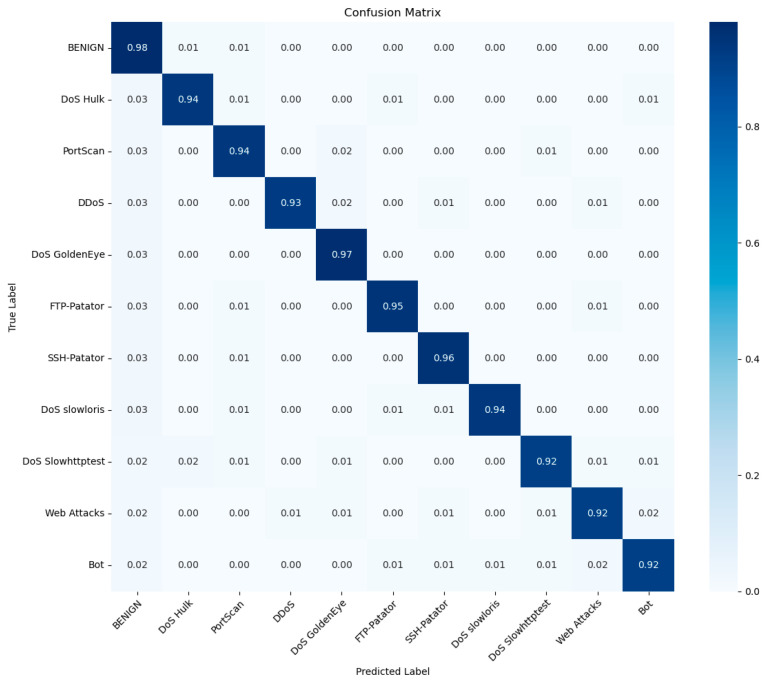
Confusion Matrix of TRBMA on Multi-Class Test Set.

**Figure 11 sensors-25-01578-f011:**
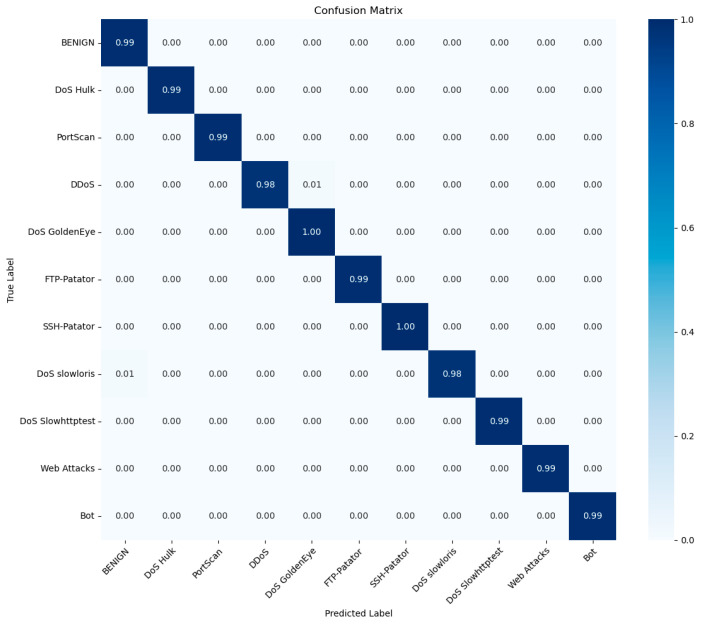
Confusion Matrix of TRBMA (BS-OSS) on Multi-Class Test Set.

**Figure 12 sensors-25-01578-f012:**
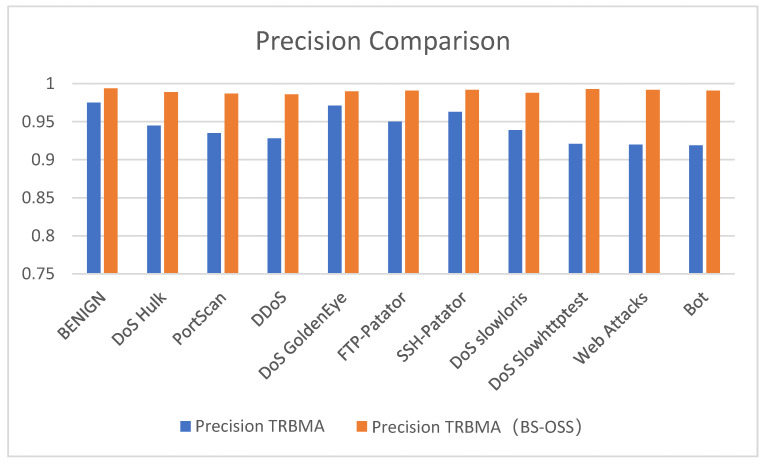
Comparison of Multi-Class Precision between TRBMA and TRBMA (BS-OSS) Models.

**Figure 13 sensors-25-01578-f013:**
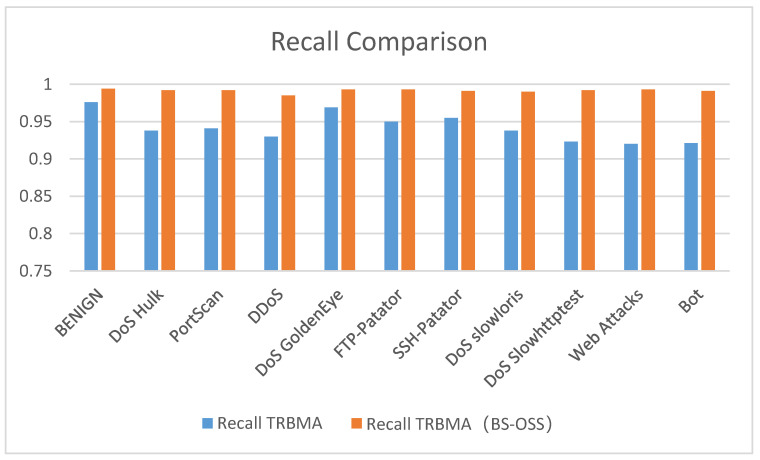
Comparison of Multi-Class Recall between TRBMA and TRBMA (BS-OSS) Models.

**Figure 14 sensors-25-01578-f014:**
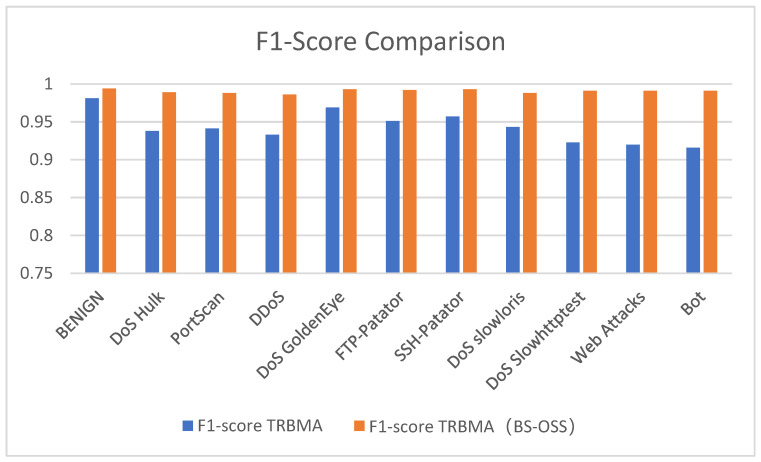
Comparison of Multi-Class F1-Score between TRBMA and TRBMA (BS-OSS) Models.

**Figure 15 sensors-25-01578-f015:**
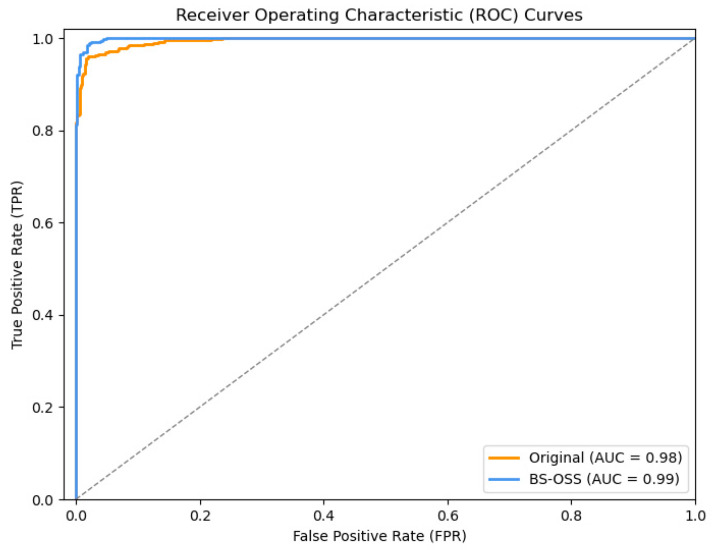
Comparison of ROC curves between TRBMA and TRBMA(BS-OSS).

**Table 1 sensors-25-01578-t001:** CIC-IDS-2017 Dataset Traffic Category Distribution.

Sub-Dataset	Network Traffic Category
Monday Samples	BENIGN
Tuesday Samples	BENIGN, FTP-Patator, SSH-Patator
Wednesday Samples	BENIGN, DoS Hulk, DoS GoldenEye, DoS slowloris, DoS Slowhttptest, Heartbleed
Thursday Morning Samples	BENIGN, Web Attack-Brute Force, Web Attack-XSS, Web Attack-Sql Injection
Thursday Afternoon Samples	BENIGN, Infiltration
Friday Morning Samples	BENIGN, Bot
Friday Afternoon Samples-DDoS	BENIGN, DDoS
Friday Afternoon Samples-PortScan	BENIGN, PortScan

**Table 2 sensors-25-01578-t002:** Post-Data Cleaning CIC-IDS-2017 Dataset Traffic Category Distribution Statistics Table.

Attack Categories	Count	Percentage (%)
BENIGN	2,273,097	80.3016997
DoS Hulk	231,073	8.1631161
PortScan	158,930	5.6145202
DDoS	128,027	4.5228099
DoS GoldenEye	10,293	0.3636208
FTP-Patator	7938	0.2804257
SSH-Patator	5897	0.2083233
DoS slowloris	5796	0.2047553
DoS Slowhttptest	5499	0.1942632
Web Attacks	2180	0.0770129
Bot	1966	0.0694529
Total	2,830,696	100

**Table 3 sensors-25-01578-t003:** Comparison of multi-category sample size between the original dataset and the training dataset based on BS-OSS hybrid sampling.

Attack Categories	Original	Hybrid Post-Sampling
BENIGN	1,818,477	108,535
DoS Hulk	184,858	107,035
PortScan	127,144	106,861
DDoS	102,421	105,712
DoS GoldenEye	8234	105,471
FTP-Patator	6350	105,356
SSH-Patator	4717	105,643
DoS slowloris	4636	105,384
DoS Slowhttptest	4399	104,572
Web Attacks	1572	103,693
Bot	1205	103,644

**Table 4 sensors-25-01578-t004:** Environment Configuration Table.

Configuration Item	Model and Version
Operating system	Windows 10
CPU	11th Gen Intel(R) Core (TM) i5-11400H @ 2.70 GHz
Memory/GB	16
Programming language	Python 3.6
Deep Learning Framework	PyTorch 1.10.2
Operating system	Windows 10

**Table 5 sensors-25-01578-t005:** Parameter Configuration Table.

Parameter	Value
Optimizer	AdamW
Batch Size	32
Epochs	100
Learning Rate	0.001
Weight	0.01

**Table 6 sensors-25-01578-t006:** Confusion Matrix.

Actual Situation	Predict Outcomes	
	Positive Examples	Negative Examples
Positive examples	TP	FN
Negative examples	FP	TN

**Table 7 sensors-25-01578-t007:** Model Performance Comparison Table.

Model	Accuracy	Precision	Recall	F1
CNN	87.52%	82.11%	87.52%	84.25%
TCN	93.54%	93.63%	93.48%	93.11%
ResNet	95.32%	94.83%	95.32%	95.03%
BiGRU	96.64%	96.87%	96.64%	96.45%
CNN-BiGRU	97.61%	97.63%	97.61%	97.59%
TCN-ResNet	97.69%	97.71%	97.68%	97.40%
TCN-ResNet-BiGRU	98.17%	98.19%	98.13%	98.18%
TGA [48]	97.83%	97.85%	97.83%	97.57%
PSO-GA-ResNet-BiGRU [24]	99.21%	97.85%	98.17%	N/A
TBGD [25]	99.08%	N/A	N/A	99%
DMFCNN [12]	95.03%	97.26%	98.75%	98.03%
PGDOFLN [50]	95%	100%	95%	97%
Model of this paper(TRBMA)	98.66%	98.70%	98.66%	98.67%

**Table 8 sensors-25-01578-t008:** Comparison of Binary and Multi-Class Classification Results of Various Hybrid Sampling Methods on TRBMA Model.

Types	Binary Classification	Multi-Classification
Unbalanced	98.26%	98.66%
SMOTE	98.17%	98.25%
ADASYN	97.97%	97.93%
Borderline SMOTE	98.64%	98.61%
SMOTE-Tomek Link	98.32%	98.78%
SMOTE-ENN	98.92%	99.29%
This paper (BS-OSS)	99.43%	99.88%

**Table 9 sensors-25-01578-t009:** Comparison of recently proposed deep learning methods based on hybrid sampling with the TRBMA (BS-OSS) method proposed in this paper.

Paper	Model	Sampling Method	ACC (%)	Pre (%)	Recall (%)	F1 (%)
[16]	RKSB	KSMOTE-Random	98.4	98.3	98.3	98.1
[35]	CNN-GRU	ADASYN-RENN	99.65	99.63	99.65	99.64
[34]	CNN	SGM(SMOTE-GMM)	99.85	N/A	N/A	99.86
[45]	ISResNet	GAN	99.86	99.57	99.91	99.74
[31]	MCBA	Borderline SMOTE-ENN	98.87	99.88	99.85	99.86
[46]	ETM-TBD	SMOTE	97.27	96.58	96.72	96.65
[15]	Transformer-CNN	ADASYN-SMOTE-ENN	99.13	99.22	99.13	99.16
[49]	Bi-LSTM	SMOTE	99.34	99.34	99.34	99.34
[18]	ResInceptNet-SA	ADASYN	99.37	99.34	99.34	99.34
[26]	SVM	WGAN	N/A	99	97	98
[51]	TMG	GAN	N/A	99.63	99.81	99.72
[52]	1DCNN	VAE-WACGAN	97.46	97.76	97.46	97.54
[53]	TACGAN	KNN	95.86	96.85	94.79	95.81
[54]	HLM	-	99.63	99.26	98.99	98.72
[55]	FTG-Net-E	-	99.67	99.25	N/A	99.29
[56]	S2CGAN	-	N/A	94.37	97.95	95.10
This paper	TRBMA	BS-OSS	99.88	99.86	99.88	99.89

## Data Availability

The CICIDS2017 dataset utilized in this research is accessible to the public. It can be located at the following URL: https://www.unb.ca/cic/datasets/ids-2017.html (accessed on 28 February 2025).

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
