# Peer review of "Research on Network Intrusion Detection Model Based on Hybrid Sampling and Deep Learning"

_sensors, 2025, doi:10.3390/s25051578_

Round 1

Reviewer 1 Report

Comments and Suggestions for Authors

The article is dedicated to the development of the TRBMA network intrusion detection model, which combines hybrid sampling methods (Borderline SMOTE-OSS) and deep learning techniques (1D-TCN-ResNet-BiGRU-Multi-Head Attention) to improve the classification accuracy of malicious traffic, particularly for attack types with a small number of samples. The study demonstrates the superiority of the proposed model compared to existing approaches using the CIC-IDS-2017 dataset. The topic of the article is relevant. However, the structure of the paper does not conform to the standard format accepted by MDPI for research articles (Introduction, including related work analysis; Models and Methods; Results; Discussion; Conclusions). The level of English is acceptable, and the paper is easy to read. The figures are of acceptable quality. The article cites 43 relevant sources.

The following remarks and recommendations can be formulated regarding the article:

1. The paper presents a combination of well-known methods (TCN, ResNet, BiGRU, Multi-Head Attention, AdamW) and their adaptation to the intrusion detection task. However, it lacks a conceptually novel approach—the proposed method is based on a combination of existing algorithms without introducing a fundamentally new architecture or methodology. The use of hybrid sampling (Borderline SMOTE-OSS) appears to be merely a combination of previously known methods (Borderline SMOTE and OSS), which limits the degree of scientific novelty.

2. The article lacks a rigorous analytical justification for the choice of methods and their combination. For example, no mathematical proof is provided to demonstrate that the proposed TRBMA architecture indeed ensures better generalization and minimizes overfitting. The architecture description is accompanied by diagrams and empirical formulas, but without a theoretical rationale for selecting these architectural decisions.

3. The "Introduction" section mentions the problem of data imbalance and the necessity of addressing it. However, the experiments do not provide an analysis of the impact of different balancing methods on model performance. In the "Results" section, it is stated that TRBMA outperforms alternative methods, but the tables show that the difference in accuracy is only a fraction of a percentage point, which may be statistically insignificant.

4. The article does not provide information on the number of experimental runs and their variability. There is no analysis of the statistical robustness of the obtained results. The experimental setup does not consider alternative datasets, making it impossible to assess the model's generalization beyond CIC-IDS-2017. Additionally, no experiment with real network traffic is conducted, which limits the practical applicability of the proposed solution.

5. The paper lacks an analysis of confidence intervals, standard deviations, and other statistical indicators confirming the stability of the results. While the authors use Precision, Recall, and F1-score, they do not consider metrics such as AUC-ROC, which could provide a more comprehensive evaluation of classification performance. Furthermore, a comparative analysis with other data balancing methods, such as ADASYN or GAN-based oversampling, is not conducted.

6. The article does not discuss computational resource requirements, which is crucial for real-world deployment in network security environments. There is no information on testing the model in real-time or its ability to operate in an online mode. Aspects of integrating the model into existing monitoring and security systems are not considered.

7. The "Related Works" section reviews numerous existing methods but does not provide a systematic comparison with the proposed model, making it difficult to assess TRBMA's advantages. The article does not sufficiently explain why the chosen architecture yields the best results. For instance, it does not explore the impact of layer depth, window size, or activation parameters. The conclusions emphasize performance improvements but do not discuss potential drawbacks, such as the model’s resilience to evolving cyberattacks.

8. Some sections are overloaded with technical details (e.g., the description of Multi-Head Attention), while other aspects (e.g., result interpretation) remain superficial. In the "Results" section, information is presented unevenly: comparative analysis with other models is shown in a table but without a qualitative discussion of the significance of the differences. The conclusion is overly optimistic but does not include recommendations for model improvement or future research directions.

Reviewer 2 Report

Comments and Suggestions for Authors

This paper proposes an enhanced network intrusion detection model, 1D-TCN- ResNet-BiGRU-Multi-Head Attention (TRBMA). 

I have the following comments for improvement: 
1- The introduction part was too short and poor. 

2- the related work section should be a section NOT a subsection, followed by a discussion part or summary table to highlight clearly the main research gaps. 

3- the references within the article should be hyperlinked to the cited ref within the ref section. 

4- the positions of the figures were incorrectly posted within the paper, for example, Figure 1.

5-'The procedural steps of the BS-OSS algorithm are outlined as follows' should be presented as algorithm instead of steps. 

6- Table 8. should include more studies. 

7- Why the CIC-IDS-2017 dataset? why not other datasets

8- Comparing your model with other techniques that utilize SMOTE or GAN should be mentioned, 

Good Luck

Round 2

Reviewer 1 Report

Comments and Suggestions for Authors

Dear Authors,

Below I provide my commentary on the revisions you have made.

In the first paragraph, I noted that the article conceptually employs a combination of well-known models (TCN, ResNet, BiGRU, attention) and suggested placing greater emphasis on scientific novelty. In the revised text, you supplemented the sections describing the parallel dual-channel architecture (TRBMA) and the extended hybrid sampling method (BS-OSS), highlighting that it is specifically optimised for network attack detection. Your clarifications lend more credibility and underscore the distinctive value of your integrated approach, although there remains an impression that the paper primarily focuses on engineering enhancements within known methodologies.

In the second paragraph, I requested more in-depth theoretical justification for the architectural choices. You indicated that you have expanded Sections 3.2–3.5 with formal descriptions of ResNet, TCN, and BiGRU, and also included experimental comparisons in Section 5.1. Indeed, additional formulas, references to ablation studies, and discussions of why this particular scheme (TCN + 1D-ResNet + BiGRU) is well-suited to time series have appeared. However, the strictly mathematical foundation is still presented in a moderate way, principally through experimental outcomes and the description of key blocks.

In the third paragraph, I pointed out insufficient evaluation of each balancing method’s contribution. You included results for SMOTE, Borderline SMOTE, ADASYN, and OSS, showing how they affect confusion matrix granularity and F1-scores for underrepresented classes. Although only a small difference is observed in the overall accuracy, the improvement is notably more pronounced for rare attack types. This bolsters the argument in favour of the BS-OSS hybrid approach.

In the fourth paragraph, I recommended conducting a statistical analysis of repeated runs and assessing variability, as well as considering real data beyond the CIC-IDS-2017 dataset. The revised text presents several runs (mean values and deviations), partially making up for the absence of confidence interval analyses. You justify the applicability to real-world traffic by noting that CIC-IDS-2017 itself is compiled from real segments. Even so, there is still no testing on other datasets or in a production environment. For a first iteration, this may be sufficient, but in future, a broader scope would be desirable.

The fifth remark concerned the use of ROC/AUC and a more detailed analysis of result stability. Your response indicates that you have added learning curves, calculations of mean and standard deviations for accuracy, as well as comparisons with methods beyond SMOTE (e.g. ADASYN). These clarifications provide a more robust set of metrics, though mentions of AUC-ROC remain somewhat brief. Nonetheless, the comparative analysis carried out can be regarded as a step forward.

The sixth point concerned computational overhead and practical application. You noted that training on an RTX 3090 took 4.2 hours for 100 epochs, and that in prediction mode the average processing time per traffic sample is around 2.3 milliseconds, describing possibilities such as pruning and quantisation in future. This is a useful addition, although questions about real-time deployment and integration into industrial systems largely remain in the realm of future plans.

The seventh remark highlighted a lack of systematic comparisons with alternative architectures. In response, you expanded the comparative analysis section (mentioning TGA, PSO-GA-ResNet-BiGRU, DMFCNN, PGDOFLN, etc.), provided tabular results, and explained why the new model achieves higher F1-scores for rare attacks. These additions indeed help elucidate the advantages and limitations of your method.

Finally, the eighth point concerned imbalanced presentation: some sections were overloaded with technical details, whereas the conclusions were overly optimistic and did not include a clear mention of potential improvements. Based on the revised version, you have added more extensive commentary on experimental results and listed plans for FPGA integration and the application of broader datasets (UNSW-NB15, IoT-23). These are helpful additions that temper excessive optimism and give readers greater insight into how the article fits into the broader research landscape.

Overall, the revisions you have introduced substantially improve the manuscript. It has become clearer how exactly TRBMA (combined with BS-OSS) stands out among existing methods, and the analysis has been expanded for classes with limited numbers of samples. Nonetheless, certain aspects may require further development in the future (for example, a more detailed mathematically grounded analysis, an independent test on another real dataset or in an industrial environment, and inclusion of more extensive AUC-ROC metrics). Nevertheless, at this stage, the revised version is more convincing than the original manuscript.

Author Response

Thank you for your thoughtful feedback and constructive criticism on our manuscript. We wholeheartedly agree with your suggestions for future work, including independent testing in another real-world dataset or industrial environment, as well as the incorporation of broader AUC-ROC metrics. We sincerely appreciate your acknowledgment of the improvements in the revised version.

In response to your request for a more rigorous theoretical justification, we have further refined the paper. We are in section 3.2 on pages 8 and 9 The architecture of the TRBMA model provides a more detailed and mathematically based theoretical justification for the TRBMA model (lines 319-323, 330-335, 338, 343-346, 380-387). And mathematical analysis of the model structure was added on lines 419 to 425 on page 10, lines 441 to 456 on page 11, and lines 508 to 514 on page 13, with a certain number of references cited as theoretical support. Finally, in section 5.1 On pages 19 to 20 of the Comparative Analysis of Models, lines 682 to 684, 687 to 691, 693 to 698, and 709 to 716, additional analyses of the experiments have been added. These analyses are based on the mathematical aspects that have been supplemented, forming a mutual correspondence and verification with the theoretical argument section mentioned earlier.

Reviewer 2 Report

Comments and Suggestions for Authors

The authors addressed my concerns. 

Good Luck

Author Response

Thank you sincerely for your constructive feedback and for acknowledging our efforts in addressing your concerns. We are delighted to hear that the revisions have met your expectations. Should any further clarifications or adjustments be required, please do not hesitate to let us know. We deeply appreciate your time and expertise in guiding the improvement of this work.